# Atrial Cardiomyopathy in Valvular Heart Disease: From Molecular Biology to Clinical Perspectives

**DOI:** 10.3390/cells12131796

**Published:** 2023-07-06

**Authors:** Andrea Ágnes Molnár, Attila Sánta, Dorottya Tímea Pásztor, Béla Merkely

**Affiliations:** Heart and Vascular Center, Semmelweis University, 1122 Budapest, Hungary

**Keywords:** atrial cardiomyopathy, fibrosis, mitral valve regurgitation, aortic valve stenosis

## Abstract

This review discusses the evolving topic of atrial cardiomyopathy concerning valvular heart disease. The pathogenesis of atrial cardiomyopathy involves multiple factors, such as valvular disease leading to atrial structural and functional remodeling due to pressure and volume overload. Atrial enlargement and dysfunction can trigger atrial tachyarrhythmia. The complex interaction between valvular disease and atrial cardiomyopathy creates a vicious cycle of aggravating atrial enlargement, dysfunction, and valvular disease severity. Furthermore, atrial remodeling and arrhythmia can predispose to atrial thrombus formation and stroke. The underlying pathomechanism of atrial myopathy involves molecular, cellular, and subcellular alterations resulting in chronic inflammation, atrial fibrosis, and electrophysiological changes. Atrial dysfunction has emerged as an essential determinant of outcomes in valvular disease and heart failure. Despite its predictive value, the detection of atrial fibrosis and dysfunction is challenging and is not included in the clinical routine. Transthoracic echocardiography and cardiac magnetic resonance imaging are the main diagnostic tools for atrial cardiomyopathy. Recently published data have revealed that both left atrial volumes and functional parameters are independent predictors of cardiovascular events in valvular disease. The integration of atrial function assessment in clinical practice might help in early cardiovascular risk estimation, promoting early therapeutic intervention in valvular disease.

## 1. Introduction

The concept of atrial cardiomyopathy was defined in 2016; however, it is still not profoundly characterized in terms of pathophysiology, risk factors, stages, progression, and prognostic value [1]. Atrial cardiomyopathy was thought to be only a secondary consequence of cardiac diseases, such as valvular heart disease. Pressure overload caused by aortic and mitral stenosis, or volume overload due to mitral and tricuspid regurgitation may lead to atrial cardiomyopathy including functional, structural, or electrophysiological remodeling [2] (Figure 1). Two main etiological entities of valve regurgitation have been defined: organic and functional. Functional valvular regurgitation was believed to be mainly a consequence of ventricular dysfunction. Recently, the entity of atrial functional valvular regurgitation has been separated from the ventricular functional one based on the pathomechanism and etiology [3]. Atrial cardiomyopathy due to chronic atrial fibrillation may lead to pronounced mitral or tricuspid annular dilatation inducing valve malcoaptation, and, consequently, atrial functional regurgitation [3,4] (Figure 1). The differentiation between ventricular functional valve disease and atrial functional valve disease is important because of the different management and prognostic implications [3]. Clinical diagnostic tools, such as imaging, serum, and electrophysiological biomarkers, are used in daily practice. It is recognized that atrial cardiomyopathy is progressive, although the stages of severity have not yet been defined. Nonetheless, the severity of atrial cardiomyopathy is inversely related to the sensitivity of its detection [1]. Furthermore, its pathophysiological evaluation has become an emerging bench-to-bedside research area involving oxidative stress, inflammation, calcium overload, profibrotic signaling pathways, and microRNA-induced pathways [5]. Despite advances, the pathogenesis and clinical impact of atrial cardiomyopathy are only partly known [6]. This review summarizes the most important clinical features of atrial cardiomyopathy in valvular heart disease and highlights the most significant cellular, molecular, and neurohormonal factors underlying atrial remodeling.

## 2. Physiology and Pathophysiology of Atria: The Concept of Atrial Cardiomyopathy

The atrial myocardium consists of two thin layers of muscle. The atrial cardiomyocytes are arranged in well-organized muscle bundles. Between the muscle bundles extends the thin atrial interstitium, which is composed mainly of an extracellular matrix of collagen type I fibers, which account for 5% of the atrial wall volume. The main cellular components in the interstitium are fibroblasts, adipocytes, mesenchymal cells, and inflammatory cells [7] (Figure 2). The cardiomyocytes are mainly mononuclear cells with perinuclear granules containing cardiac hormones, such as atrial natriuretic peptide (ANP) and brain natriuretic peptide (BNP), which are secreted into the circulation in response to the dynamics of cardiac pressure and volume [8]. The ANP is promptly released from granules in response to even minor triggers and can indicate acute hemodynamic changes [9]. In contrast, BNP has minimal granular storage, and is consequently regulated transcriptionally and synthesized and secreted intermittently [9,10]. Increased atrial pressure and wall stress induce the release of natriuretic peptides into the circulation. These peptides generate natriuresis, diuresis, vasodilation, and inhibit aldosterone synthesis and renin secretion, which enable these cardiac hormones to regulate blood pressure and volume hemodynamics [11,12]. BNP is secreted both by atrial and ventricular cardiomyocytes. A greater proportion of BNP is produced by ventricle under physiologic circumstances, due to the greater ventricular mass compared to atrial mass [13,14]. ANP is produced only by atrial cardiomyocytes; however, its superiority in atrial cardiomyopathy diagnostics is debatable [13]. These hormones bind to membrane-bound natriuretic peptide receptors of the target organs. Natriuretic peptides stimulate water excretion in the kidney through the contraction of efferent arterioles and relaxation of afferent arterioles in nephrons, enhancing glomerular filtration [12,15]. The natriuretic effect is a consequence of reduced sodium reabsorption in the renal tubules [12,15]. Furthermore, ANP and BNP act on vascular smooth muscle cells by affecting the permeability of the vascular endothelium, and increasing the fluid shift from the intravascular to the extravascular compartment [12,15]. Notably, natriuretic peptides exert both antifibrotic and antihypertrophic effects in the heart [11,16].

The increased atrial pressure in valvular heart diseases triggers mechanical stimuli in the atrial wall and initiates complex molecular, cellular, and neurohormonal interlinked pathways [6,17] (Figure 2). The mechanical atrial wall stress activates the renin–angiotensin–aldosterone system (RAAS), which promotes fibrosis via several pathways, such as the activation of hydrolysis phospholipase C (PLC), leading to intracellular Ca^2+^ overload and fibroblast proliferation [5,6] (Figure 2). Furthermore, angiotensin II activates nicotinamide adenine dinucleotide phosphate (NADPH) oxidase and the release of reactive oxygen species (ROS), which induce intracellular Ca^2+^ overload and fibrosis via the mitogen-activated protein kinase (MAPK) pathway [6,18] (Figure 2). In addition, angiotensin II regulates the expression of profibrotic factors, such as transforming growth factor β (TGF-β) and connective tissue growth factor (CTGF) [6,18] (Figure 2). Angiotensin II induces the hypertrophy of atrial cardiomyocytes, characterized by an increase in cell size and the reinduction of fetal cardiac genes via transcription factors and the epigenetic regulation of gene transcription regulation [5,19]. Atrial stretch also triggers leukocyte activation and the subsequent release of inflammatory stimuli [5,20]. Furthermore, atrial stretch and inflammation activate fibroblast proliferation and differentiation into the myofibroblast phenotype to produce extracellular matrix (ECM) components including fibronectin, procollagen, laminin, elastin, fibrillin, proteoglycans, glycoproteins matrix metalloproteinases (MMPs), and tissue inhibitors of MMPs [5,17] (Figure 2). In the “reactive” form of fibrosis (or interstitial fibrosis), the accumulation of ECM is observed mainly in the perivascular space and in the perimysium surrounding cardiac muscle bundles [17,21]. In “reparative” fibrosis, the fibrous tissue replaces apoptotic or necrotic cardiomyocytes, creating irreversible interruptions in myocardial bundles [17]. However, in many cases, a “mixed” form of interstitial and reparative fibrosis develops. Notably, fibrosis can be induced by fatty infiltration and the amyloid deposition of interstitial matrix without prominent collagen fiber accumulation, defined as “non-collagen” fibrosis [7]. The histological and pathophysiological classification of atrial cardiomyopathy has been defined by the European Heart Rhythm Association (EHRA), the Heart Rhythm Society (HRS), the Asian Pacific Heart Rhythm Society (APHRS), and Sociedad Latino Americana de Estimulacion Cardiaca y Electrofisiologia (SOLAECE) working groups, determining four EHRAS (EHRA/HRS/APHRS/SOLAECE) class groups [7]. Accordingly, EHRAS class I represents cardiomyocyte changes (“reparative” fibrosis), class II fibrotic changes (“reactive” fibrosis), class III combined cardiomyocyte pathology with fibrosis (“mixed” form), and class IV non-collagen infiltration with or without cardiomyocyte changes [7]. In addition to valvular heart disease, further risk factors, such as hypertension, diabetes mellitus, and heart failure, might contribute to fibrogenesis [7]. The presence of any type of fibrosis promotes changes in impulse propagation and may induce subsequent re-entrant atrial arrhythmias [5,17]. Furthermore, inflammation may lead to a dysfunction of cardiomyocyte ion channels, promoting arrhythmia [5]. Notably, fibroblasts can connect to cardiomyocytes via gap junctions and consequently act as passive electrical conduits; however, they are not electrically excitable [5,6,22]. A growing body of evidence supports that atrial fibrillation is associated with systemic vascular, coronary, and atrial endothelial dysfunction through multiple mechanisms [23]. Previous studies have shown that altered shear stress on endothelial cells, increased oxidative stress and inflammation, RAAS activation, intracellular Ca^2+^ overload, the release of endothelin-1 (ET-1), and reduced nitric oxide production are the main pathways leading to atrial cardiomyopathy [23,24,25,26].

Functionally, the left atrium serves as a conduit between the pulmonary circulation and left ventricle [27]. Similarly, the right atrium receives peripheral venous blood, which passes toward the right ventricle [28]. Nonetheless, atria are not only passive chambers that transfer blood from veins to ventricles. Both atria actively modulate ventricular filling by acting as reservoirs during ventricular systole, and as conduits for early ventricular diastole, and finally as booster pumps during late ventricular diastole, when the atria contract. Atrial contraction and the booster pump function are absent in atrial fibrillation. Accordingly, atrial filling occurs during pulmonary venous return in the reservoir and conduit phase, whereas ventricular filling occurs in the conduit and booster pump phase. Consequently, the reservoir function depends mainly on atrial compliance, end-systolic ventricular volume, and the descent of the ventricular base during systole; however, the conduit function is determined by both atrial and ventricular compliance. Atrial booster-pump function depends on atrial preload and afterload, represented by venous return and end-diastolic ventricular pressures, respectively [7,27,29].

## 3. Biomarkers of Atrial Cardiomyopathy: Serological, Electrophysiological, and Imaging Biomarkers

### 3.1. Serological Biomarkers in Atrial Cardiomyopathy

The hemodynamic load in valvular heart disease can lead to elevated atrial pressure and atrial cardiomyopathy, triggering the secretion of natriuretic peptides from the atrial wall. In general, valvular heart disease impacts both the atria and ventricles; therefore, natriuretic peptides usually represent a surrogate marker of overall cardiac stress in valvular heart disease, and not just atrial cardiomyopathy. The natriuretic peptide level increases with the progression of valvular heart disease and heart failure, and therefore provides important prognostic information [30]. This is also used for monitoring patients before and after valvular interventions, and to help with the risk stratification and timing of invasive treatment [30]. Current guidelines recommend the clinical use of natriuretic peptides for the diagnosis and prognosis of heart failure [31]. Concentrations of different natriuretic peptides can be measured from blood samples with similar diagnostic and prognostic accuracy, although they should never be a stand-alone test in the heart failure diagnostic work-flow [32]. Plasma levels of ANP secreted from the atria are higher than BNP plasma levels secreted from both atria and ventricles in healthy subjects [10,33,34]. However, both ANP and BNP are elevated in heart failure, when BNP plasma levels can exceed those of ANP [10,33,34]. Increased atrial and/or ventricular wall stress in valvular heart disease triggers the synthesis of pre-proBNP, which is cleaved to proBNP, and then to the biologically active BNP hormone and the inactive amino-terminal fragment, N-terminal-pro-B-type natriuretic peptide (NT-proBNP) [30]. Notably, in clinical practice, NT-proBNP is more commonly used because of its longer half-life (120 min) compared to the half-life of BNP (22 min) [35,36]. The normal cut-off values of natriuretic peptides depend on the assessment method, age, gender, and body mass index [36,37]. “Normal” reference values for NT-proBNP from the Framingham Heart Study ranged from 42.5 pg/mL to 106.4 pg/mL in men, depending on age [37]. Women showed different normal NT-proBNP cut-off values between 111.0 pg/mL and 215.9 pg/mL [37]. Recent guidelines from the European Society of Cardiology have established a threshold of 125 pg/mL NT-proBNP and 100 pg/mL BNP for symptomatic heart failure patients in a non-acute setting; however, the use of this single cut-off value might be suboptimal for screening the general population [31,32,38]. Notably, the Generation Scotland Scottish Family Health Study reported that NT-proBNP ≥ 125 pg/mL was detectable in 10% of young females without cardiovascular risk factors [38]. Furthermore, anemia, renal failure, obesity, hyperthyroidism, sepsis, and pulmonary hypertension may represent confounding factors in the interpretation of NT-proBNP levels in heart failure [30,36,39]. Besides heart valve disease, NT-proBNP has been established as a marker of atrial fibrillation and ischemic stroke [40]. Atrial fibrillation was independently associated with greater NT-proBNP elevation and left atrial volumes compared to sinus rhythm in heart failure population [40]. Other natriuretic peptides, such as ANP and mid-regional-proANP (MR-proANP), are less prevalent [32]. MR-proANP is a stable fragment of the precursor hormone ANP, has no receptor binding and protein interaction, and has a longer half-life compared to ANP [6,41,42,43]. MR-proANP and ANP are produced only by the atria and are therefore thought to be more specific biomarkers of atrial cardiomyopathy compared to BNP and NT-proBNP [6,44]. Previous studies have shown that atrial arrhythmias are associated with higher levels of BNP, NT-proBNP, ANP, or MR-proANP, even in the absence of heart failure, due to increased wall stress induced by tachycardia [44,45,46,47]. Meune et al. found that MR-proANP levels were lower; if the onset of atrial fibrillation was less than 48 h, consequently, MR-proBNP levels might identify the time from the onset of atrial fibrillation [47]. Notably, natriuretic peptide levels incrementally increase in paroxysmal and chronic atrial fibrillation compared to sinus rhythm [41]. Importantly, natriuretic peptides cannot identify the etiology of heart failure [32]. Subsequently, complementary cardiac imaging is useful in interpreting the underlying pathology in elevated cases [32]. Cui K. et al. found that plasma levels of MR-proANP were associated with left atrial volumes and functional stages of heart failure; however, NT-proBNP had no correlation [48]. Furthermore, NT-proBNP levels independently predicted the prevalence of left atrial fibrosis in a population with atrial fibrillation [6,49].

Vascular cell adhesion molecule 1 (VCAM-1) is a cell surface protein that contributes to leukocyte infiltration, inflammation, and atrial remodeling; however, it is not used in clinical practice [6]. ADAMs (A Disintegrin and Metalloproteinase) proteins might be additional potential biomarkers in atrial cardiomyopathy [50]. ADAMs are membrane-bound glycoproteins known to regulate cell–cell and cell–matrix interactions and may thereby influence cardiac remodeling [50]. Arndt M. et al. have reported that atrial fibrillation is associated with an increased expression of these proteins, which might contribute to atrial dilation in arrhythmia [50]. Previous animal studies have revealed that ADAMs may also be related to myocardial hypertrophy and fibrosis [51,52].

### 3.2. Electrophysiological Markers in Atrial Cardiomyopathy

Electrophysiological, structural, and functional atrial remodeling, including fibrosis, represent a substrate for atrial fibrillation. Notably, atrial fibrillation may stimulate further atrial remodeling, which hence maintains and aggravates the vicious cycle of atrial cardiomyopathy and atrial fibrillation [17,53,54]. Hopman et al. compared atrial function in atrial fibrillation patients with higher and lower degrees of left atrial fibrosis [55]. Higher levels of atrial fibrosis were associated with lower atrial function compared to patients with less fibrosis [55]. Atrial dysfunction might develop before atrial enlargement; therefore, it can serve as an early biomarker of cardiac injury and the risk of atrial fibrillation [27]. Furthermore, atrial fibrillation may promote atrial enlargement. The presence of valvular heart disease further complicates the pathophysiological vicious cycle of atrial cardiomyopathy, atrial fibrillation, and valvular heart disease (Figure 1).

It is recognized that atrial fibrillation and atrial cardiomyopathy are associated with a risk of atrial thrombogenesis and cardiogenic embolization [56,57]. Ineffective atrial contraction and low flow conditions promote thrombus formation, most commonly in the left atrial appendage, thereby increasing the risk of stroke and peripheral embolization [58] (Figure 1). Previous studies have revealed substantial prothrombogenic changes in atrial fibrillation disease with the expression of von Willebrand factor and adhesion molecules in the atrial endocardium [59]. Moreover, cardiovascular risk factors may further increase the expression of pro-thrombogenic and pro-inflammatory factors [57,60,61,62,63]. New data are emerging on the relationship between cardiovascular risk factors, inflammation, thrombogenesis, and atrial fibrillation [57,60,61,62,63]. Watanabe et al. have shown that C-reactive protein (CRP) levels and atrial diameter size were higher in atrial fibrillation than in sinus rhythm, which decreased after successful cardioversion [64,65]. Furthermore, CRP might be useful as an independent factor to predict the recurrence of atrial fibrillation [66,67]. Overall, thrombogenic risk assessment in non-rheumatic atrial fibrillation is based on the CHA_2_DS_2_-VASc Score (Chronic Heart Failure, Hypertension, Age ≥ 75 years, Diabetes mellitus, prior Stroke or TIA or thromboembolism, Vascular disease, Age 65–74 years, Sex category (i.e., female sex)) [68]. Notably, patients with cardiovascular risk factors (CHA_2_DS_2_-VASc Score ≥ 2) and paroxysmal (self-terminating) or persistent (lasting more than seven days or requiring cardioversion) atrial fibrillation have a similar risk of stroke compared to patients with permanent atrial fibrillation, as prothrombogenic molecular atrial alterations persist even during periods of sinus rhythm in these populations [7,69,70]. Consequently, continuous anticoagulant medication is required in all three atrial fibrillation entities if CHA_2_DS_2_-VASc Score ≥ 2 [68]. Previous studies have revealed that atrial cardiomyopathy, even in the absence of atrial fibrillation, is a risk factor for stroke and cognitive impairment [71,72,73]. The detection rate of new-onset atrial fibrillation after cryptogenic stroke ranged between 5% and 25%, depending on the heart-rhythm monitoring method [74,75,76,77]. This implies that atrial fibrillation may not be a necessary condition for stroke [72]. Besides atrial fibrillation, further electrocardiogram measurements have been reported as markers of atrial cardiomyopathy, such as a prolonged PR interval, P-wave morphology (terminal force in lead V1), paroxysmal supraventricular tachycardia, and ectopic atrial rhythm [78,79,80]. These alterations have been shown to increase the risk stroke and/or atrial fibrillation in large-scale studies [71,78,81,82,83]. Altered P-wave morphology might reflect elevated left atrial pressure, left atrial enlargement, and fibrosis [80,84].

### 3.3. Imaging Markers in Atrial Cardiomyopathy

#### 3.3.1. Transthoracic Echocardiography

Current recommendations suggest conventional two-dimensional (2D) transthoracic echocardiography to measure atrial dimensions and volumes which have prognostic value [85,86]. The normal upper limit of left atrial volume is 34 mL/m^2^ for both genders; meanwhile, the normal upper limit of right atrial volume is 25 ± 7 mL/m^2^ in men and 21 ± 6 mL/m^2^ in women [85]. However, 2D echocardiography is based on geometric assumptions and the foreshortening of the left atrial cavity [87]. Overall, 2D echocardiography underestimates left atrial volumes compared to cardiac magnetic resonance values [88]. Left atrial enlargement has been considered a marker of atrial cardiomyopathy and a risk factor for atrial fibrillation, stroke, heart failure, and cardiovascular mortality [72,89,90]. The presence of spontaneous echocardiographic contrast in the left atrium and reduced left atrial appendage blood flow velocity have been recognized as markers of atrial cardiomyopathy and associated with an increased risk of thrombus formation [80,91,92]. Furthermore, transthoracic and transesophageal echocardiography can easily detect deformations of the interatrial septum, such as interatrial septal aneurysm (IASA), patent foramen ovale (PFO), or atrial septal pouch (ASP) [93,94,95]. The IASA is defined as a saccular deformation of the atrial septum, the PFO is a flaplike opening between the atrial septum primum and secundum at the location of the fossa ovalis, and the diverticulum-like ASP occurs when the septum primum and secundum are incompletely fused [93,96]. These deformations are supposed to be associated with thromboembolic events in different mechanisms [93]. Local thrombus formation related to blood stasis within these deformations might represent one mechanism [93]. However, thromboembolism transit through the PFO channel might serve as a further potential mechanism, defined as a paradoxical embolism [93]. Notably, the prevalence of PFO is approximately 25% in the healthy adult population; nonetheless, the risk of embolization in this population is mainly associated with several clinical and morphological factors, such as the length of the PFO channel, the presence of concomitant IASA or a prominent Eustachian valve, and absence of cardiovascular risk factors [93,97,98].

Advanced echocardiography methods, such as speckle tracking echocardiography, represent a sensitive tool to assess atrial function. This technique analyses the unique fingerprint-like natural acoustic marker pattern (or speckle pattern) of grayscale B-mode images of the myocardium, which are tracked consecutively frame-by-frame during the cardiac cycle [99]. Atrial strain is a measurement of deformation, expressed as a fractional change in length from its original dimension in the tangential direction referred to as atrial longitudinal strain. The strain rate represents the speed at which the deformation (strain) of the myocardium occurs [29,100]. The left atrial longitudinal strain curve consists of the three atrial phases defined as reservoir, conduit, and contraction strain, which are modulated by both loading conditions and heart rate [100] (Figure 3). Notably, atrial reservoir strain is considered equal to peak atrial longitudinal strain (PALS) due to the negligible circumferential and radial strain of the thin atrial wall [29]. Similarly, conduit strain is also defined as longitudinal atrial conduit strain (LACS) and booster pump strain as peak atrial contraction strain (PACS) [29] (Figure 3). The EACVI/ASE/Industry Taskforce consensus document recommended a standardized atrial strain analysis that allows both R–R and P–P ECG wave gating as the zero reference point for the atrial strain curve [100]. The two methods are not interchangeable as the atrial length is different at the zero point and consequently results in different strain values [29,100]. This makes it difficult to compare the results of studies using different methods; however, most commonly, R–R ECG gating has been used in the literature [29]. The Copenhagen City Heart Study defined the normative absolute values of median PALS, LACS, and PACS parameters in 1641 healthy participants as being 39.4%, 23.7%, and 15.5%, respectively [101] (Figure 3). Similar normal values were published in a meta-analysis of 2542 healthy individuals [102]. There is a growing body of evidence that normal reservoir strain values become lower with age, which is compensated for by the increasing atrial contraction strain [101,103,104,105]. In healthy individuals over 65 years of age, Nielsen A.B. et al. have shown that normal PALS, LACS, and PACS absolute values are 33.5%, 15.6%, and 18.7%, respectively [101]. There are conflicting data on the effect of gender on atrial strain values [103,105]. The NORRE (Normal Reference Ranges for Echocardiography) multicenter study found no gender differences; however, several working groups reported a tendency of lower values for PALS and LACS in males [101,103,105].

#### 3.3.2. Cardiac Computed Tomography

Cardiac computed tomography (CCT) is an accurate tool to assess atrial volumes and dimensions [106]. However, CCT tends to overestimate maximal and minimal left atrial volumes compared to cardiac magnetic resonance (CMR) [88]. The Danish Cardiovascular Screening Trial (DANCAVAS) used a low-dose non-contrast CT scan to demonstrate the complex pathophysiology behind atrial cardiomyopathy due to an unexpected correlation between cardiovascular risk factors and atrial size [107]. Age, hypertension, pulse pressure, and blood pressure were associated with an increase in the left atrial area; however, smoking, diabetes, and dyslipidemia were associated with a decrease in left atrial size [107]. Furthermore, non-contrast CT-derived left atrial area was an important predictor of atrial fibrillation, heart failure, and death [108]. Although feature tracking-CT allows for atrial strain analysis, it is mainly used in research. Data are inconsistent for the correlation between CT-derived left atrial strain measurements and speckle tracking echocardiography-derived strain parameters [109,110].

#### 3.3.3. Cardiac Magnetic Resonance

Cardiac magnetic resonance is frequently used for ventricular fibrosis imaging; however, the detection of atrial fibrosis is still challenging due to the suboptimal image resolution of the thin-walled atrium (2–4 mm). Beyond the cost and availability of CMR, the time and expertise required for image processing are not negligible [86]. Late-gadolinium-enhanced (LGE) CMR and post-contrast T1 mapping are used to detect and quantify atrial fibrosis [111,112,113,114]. The Utah Staging System proposed by Oakes R.S. et al. was used to grade atrial fibrosis based on the extent of LGE: Utah stage I (<5%), stage II (5–20%), stage III (21–35%), and stage IV (>35%) [111]. Oakes R.S. et al. reported a 14% atrial fibrillation recurrence after ablation with low scar burden and 75% with high scar burden [111]. The multicenter, prospective DECAAF Study (Delayed-Enhancement MRI Determinant of Successful Radiofrequency Catheter Ablation of Atrial Fibrillation) also defined four stages of atrial fibrosis in the atrial fibrillation population using a delayed-enhancement MRI based on the volumetric percentage of left atrial wall enhancement: stage 1 (<10% of the atrial wall), 2 (between 10% and 19%), 3 (between 20% and 29%), and 4 (≥30%) [113]. For stage 1, the incidence of recurrent arrhythmia in the DECAAF study following successful radiofrequency catheter ablation by day 325 was similar (15.3%) to the results of Oakes et al. [111,113]. A higher incidence of atrial fibrillation recurrence (51.1%) was found in stage 4 in the DECAAF study, suggesting an independent association between atrial fibrosis stage and recurrent arrhythmia [113]. Interestingly, among patients with persistent atrial fibrillation in the DEACAAF II trial, MRI-guided atrial fibrosis ablation with pulmonary vein isolation was not superior to conventional pulmonary vein isolation alone in reducing the recurrence of atrial fibrillation [115]. Furthermore, novel atrial four-dimensional (4D) flow CMR-derived atrial flow dynamics and stasis quantification might reveal atrial predisposition sites to thrombogenesis, even with a low CHA2DS2-VASc score [114,116]. The CMR feature tracking (CMR-FT) method may provide a more accurate tool for assessing atrial strain and strain rate due to higher tracking quality compared to speckle tracking echocardiography. However, its clinical application is still limited [86,117]. The normal mean values for the reservoir, conduit, and booster strain using CMR-FT are 39.13% ± 9.27, 25.15% ± 8.34, and 13.99% ± 4.11, respectively [118]. The conduit function gradually decreases with age, while the booster atrial function significantly increases [118]. However, no significant gender differences could be revealed [118].

## 4. Valvular Heart Disease and Atrial Cardiomyopathy

### 4.1. Aortic Valve Stenosis

Aortic valve stenosis is the most common valvular heart disease, especially in the elderly population [119]. Left ventricular pressure increases in aortic stenosis leading to adaptive cardiomyocyte hypertrophy and concentric left ventricular hypertrophy. This adaptive response decreases wall stress and maintains normal left ventricular function [120,121]. However, left ventricular diastolic function deteriorates over time and left ventricular filling pressures and atrial pressures increase, leading to left atrial dysfunction and enlargement [122]. In an advanced decompensated condition, additional left ventricular systolic dysfunction develops due to increased myocardial oxygen consumption and decreased oxygen supply leading to ischemia and fibrosis [120,121]. A higher atrial pressure induces mechanical stress on the atrial wall, which generates fibroblast activation and atrial fibrosis [123] (Figure 2). Left atrial dysfunction might appear before atrial enlargement and ventricular damage [124,125]. This could be an initial indicator of atrial cardiomyopathy and might reveal the presence of cardiac injury in aortic stenosis [124,125]. Atrial remodeling in aortic stenosis can serve as a substrate for atrial fibrillation [126,127] (Figure 1). Previous studies have shown that the prevalence of atrial fibrillation in a population with aortic stenosis is approximately 10–13% and can reach even 50% in the symptomatic severe aortic stenosis group [2,126,127]. Importantly, it has been recognized that left atrial dysfunction precedes dilatation. Furthermore, left atrial size increases with the severity of aortic stenosis reflecting the chronicity of increased left ventricular filling pressure [128]. In addition, recent publications have shown that PALS is impaired in aortic stenosis and is associated with left ventricular filling pressures and left atrial fibrosis [122,124,129,130]. Overall, a large body of literature has demonstrated that left atrial strain is the predominant prognostic factor in aortic stenosis over conventional echocardiographic parameters including left atrial volume [122,124,129,130]. Tan E.S.J. et al. found that PALS < 20%, LACS < 6%, and PACS < 12% identified patients at a higher risk of adverse outcomes [129]. Similarly, Galli et al. found that PALS < 21% is an independent predictor of prognosis in aortic stenosis [130]. Furthermore, the working group suggested that reduced PALS might be a biomarker of global myocardial impairment [130]. BNP and NT-proBNP have been associated with the severity of aortic stenosis and heart failure functional status [131,132]. Notably, even moderate aortic stenosis is associated with a higher risk of mortality compared to the general population, which is mainly related to associated comorbidities, such as atrial fibrillation [133]. There is growing evidence that NT-proBNP, left atrial reservoir function, and volume are among the main prognostic markers in asymptomatic severe and moderate aortic stenosis, beyond the conventional clinical and echocardiography indices [133,134,135]. Bergler-Klein et al. have shown that asymptomatic patients with severe aortic stenosis are unlikely to develop symptoms within 6 months at low plasma natriuretic peptide levels [134].

Current guidelines recommend aortic valve replacement to treat severe aortic valve stenosis, as no medical treatment is available [29,119]. Besides surgical valve replacement, transcatheter aortic valve replacement (TAVR) has emerged as an effective treatment for patients at a high or intermediate surgical risk [119,136,137]. The cessation of left ventricular pressure afterload was associated with a 47% reduction in left atrial size shortly after TAVR (a median follow-up time of 7 days) [138]. These patients had a better improvement in clinical status and lower rates of major adverse cardiac events at one-year follow up, than patients with unchanged left atrial size [138]. The vast majority of literature reported an improvement in left atrial reservoir function following aortic valve replacement [139,140,141,142,143,144]. Interestingly, most studies did not exclude confounding factors of atrial strain assessment, such as atrial fibrillation and/or moderate-severe mitral valve disease [139,142,143,145]. Weber et al. reported that left atrial reservoir function and volume, in addition to left ventricular function, were associated with worse outcomes after TAVR [141]. Sabatino et al. found no significant improvement in atrial function and size in a large cohort of patients following TAVR at a median follow-up time of 31 months [122]. The study group showed that the lack of improvement in left atrial reservoir strain after TAVR was associated with a worse outcome [122]. Left atrial contraction strain was less frequently examined in the studies, but most studies found an improvement in left atrial contraction function [139,140,144,146]. The incidence of postoperative atrial arrhythmias requiring treatment after aortic valve replacement was 48.8%. This incidence is higher (60.1%) if the aortic valve replacement is combined with coronary artery bypass grafting [147]. The PARTNER (Placement of Aortic Transcatheter Valve) 3 trial revealed that patients undergoing aortic valve replacement and developing post-discharge atrial fibrillation had worse outcomes, irrespective of the valve replacement technique (surgical or transcatheter) [148]. Cameli et al. reported that preoperative atrial dysfunction with a cut-off value of 16.9% for PALS was associated with a risk of postoperative atrial fibrillation after surgical valve replacement [149]. Large-scale studies demonstrated that at least one-third of TAVR patients had poor outcomes as defined by mortality and quality of life, mainly attributed to atrial fibrillation, stroke, age, male gender, diabetes, and severe renal and pulmonary dysfunction [150,151]. This suggests that the current timing of aortic valve replacement may be suboptimal, as irreversible maladaptive changes might have already developed by the time of intervention.

### 4.2. Mitral Valve Regurgitation

Mitral regurgitation is the second most common valvular heart disease with a complex pathophysiology [152]. Diseases of the mitral valve apparatus (leaflets, chordae tendineae, papillary muscles, and annulus) lead to primary (organic) mitral regurgitation (Figure 4). Mitral annulus dilatation and leaflet malcoaptation due to left ventricular or atrial dilatation are defined as ventricular and atrial functional (secondary) mitral regurgitation, respectively [153,154]. Atrial functional mitral regurgitation is a unique form that initially presents with normal left ventricular function and volumes, but with a dilated mitral annulus due to left atrial enlargement as a consequence of persistent atrial fibrillation and/or heart failure with preserved ejection fraction (HFpEF) [4,153,154,155] (Figure 1). Ventricular functional mitral regurgitation is the consequence of ischemic or non-ischemic (e.g., dilated) cardiomyopathy. Nevertheless, both in organic and functional etiologies, blood from the left ventricle is ejected forward into the aorta and backward into the left atrium, resulting in an elevated total stroke volume [153,154]. Volume overload leads to left atrial dysfunction, which can be used as an early sign of myocardial damage before left ventricular dysfunction occurs [156]. Left ventricular subclinical damage develops only years after the onset of chronic volume overload and is detected by left ventricular global longitudinal strain [156]. Left ventricular and atrial dilatation occurs as an additional step to reduce wall stress and maintain normal intracardiac pressures, also defined as asymptomatic compensatory eccentric left ventricular hypertrophy. At this stage, the reversibility of cardiac damage is already uncertain [153,154,156]. Furthermore, long-standing chronic volume overload can result in progressive left ventricular and atrial enlargement and a reduction in conventional left ventricular ejection fraction, leading to a symptomatic decompensated condition with increased left atrial and ventricular diastolic pressures and an increase in pulmonary vascular resistance [153,154]. Finally, biventricular dysfunction is detected usually at a late stage, with an uncertain beneficial effect from invasive treatment [156].

Several studies have investigated the cellular and molecular pathophysiology of atrial cardiomyopathy in mitral regurgitation. Atrial volume and pressure overload result in atrial myocardial overstretching and increased myocyte oxidative stress [157,158]. Chang et al. found that atrial stretch in mitral regurgitation triggers NADPH oxidase activity and superoxide production, which impairs cellular energetic homeostasis [157,159]. Consequently, this activates the programmed cell death of myocytes independently of atrial fibrillation [157,160]. Hemodynamic load and increased atrial myocardial stretch may trigger the adaptive dedifferentiation of atrial cardiomyocytes and fibroblast proliferation, which induces the further dedifferentiation of cardiomyocytes independent of atrial fibrillation [161]. Dedifferentiated atrial cardiomyocytes are characterized by the expression of the fetal α-smooth muscle actin isoform and low numbers of sarcomeres with limited contractile capacity [161,162]. Nevertheless, fetal cardiomyocytes are considered to be stable-surviving cells due to having small mitochondria and consequently lower oxygen requirements [161,162]. Atrial-stretch-induced fibroblast activation and proliferation lead to interstitial fibrosis and replace apoptotic cardiomyocytes [123,161] (Figure 2). Chen et al. found similar amounts of atrial interstitial fibrosis in both sinus rhythm and atrial fibrillation in mitral and tricuspid regurgitation [161]. This suggests that atrial dysfunction in mitral regurgitation is primarily due to fibrosis caused by wall overstretch and the apoptosis or dedifferentiation of atrial cardiomyocytes [161]. Cameli et al. have shown that left atrial dysfunction correlates with the severity of mitral regurgitation; the more severe the mitral regurgitation, the lower the left atrial reservoir strain [163]. In addition, plasma levels of natriuretic peptides increase with the severity of mitral regurgitation and serve as a prognostic marker [164,165]. Furthermore, the amount of atrial fibrosis correlates with the atrial reservoir strain [166]. Notably, the restoration of sinus rhythm in atrial functional mitral regurgitation might be beneficial through reverse left atrial remodeling [167]. However, the duration of atrial fibrillation is inversely correlated with postcardioversion or postablation sinus rhythm permanence [168]. Hence, this suggests that sinus rhythm restoration should be performed in the early stage of atrial fibrillation disease.

Surgery and transcatheter interventions are invasive treatment options for severe mitral regurgitation [119]. The cessation of or reduction in regurgitant volume after intervention reduce left atrial extension and pressure; however, the improvement in left atrial function might depend on preoperative strain values [169]. Improvement in atrial function after transcatheter mitral valve repair (e.g., MitraClip implantation) is controversial [169,170,171,172,173,174,175]. Some study groups have found improvements in left atrial function after the transcatheter procedure; however, others have found no relevant change or even worsening [169,170,171,172,173,174,175]. This might be due to the low study sample size and differences between study designs, including the etiology of mitral regurgitation, left ventricular function, the presence of atrial fibrillation, and the severity of residual mitral regurgitation after MitraClip implantation [169,170,171,172,173,174,175]. Avenatti et al. found an improvement in atrial function after MitraClip intervention, regardless of etiology [173]. In surgical mitral valve repair, preoperative left atrial strain values and volumes are independent predictors of new-onset atrial fibrillation after surgery for mitral valve regurgitation [176]. In addition, left atrial reservoir strain, with a cut-off value of 22%, was associated with all-cause mortality after mitral valve repair for primary mitral regurgitation, and provided an incremental prognostic value over left atrial volume [177]. Nonetheless, current guidelines recommend surgical treatment for severe primary mitral regurgitation in the presence of either symptoms or left ventricular dilatation and dysfunction, high pulmonary pressure, left atrial dilatation, and/or atrial fibrillation [119]. Notably, an assessment of left atrial function in primary mitral regurgitation has been shown to have a superior prognostic value over established surgical indicator parameters [156,178,179,180] (Figure 4). Although some authors have suggested the addition of left atrial strain to established indices indicating mitral valve surgery, the use of early signs of atrial cardiomyopathy, such as atrial strains in decision making, remains a question of further investigation [180]. Importantly, atrial dysfunction is not comparable in MitraClip and surgical mitral valve repair population, as generally there is no residual mitral regurgitation following surgical mitral valve repair; however, mild-to-moderate residual mitral regurgitation is common after MitraClip implantation [169]. Furthermore, comorbidities affecting atrial function can be different in surgical and transcatheter mitral valve repair populations [169]. Left ventricular function is an important contributor of left atrial reservoir strain, which is reduced only in ventricular functional regurgitation, usually suitable only for MitraClip implantation because of higher surgical risk [169,181,182]. In addition, the impact of atriotomy on left atrial function in the case of surgery cannot be neglected [169]. Overall, the optimal timing of mitral valve surgery is of utmost importance; therefore, the early detection of signs of myocardial damage is crucial [183].

### 4.3. Aortic Valve Regurgitation

Aortic valve regurgitation occurs when the aortic valve does not close tightly due to a disease of the valve cusp and/or a dilatation of the aortic root. This leads to the diastolic backflow of blood from the aorta into the left ventricle, leading to left ventricular volume and pressure overload, mainly in severe conditions. The systolic function of the left ventricle is preserved and filling pressure is normal in a compensated state due to adaptive left ventricular eccentric hypertrophy caused by lengthened myofibrils [184]. In advanced stages, interstitial fibrosis leads to reduced left ventricular compliance and function, leading to high filling pressures and symptoms of heart failure [185]. Increased left atrial pressure affects left atrial mechanics, and pulmonary hypertension might develop. Kalkan S. et al. found that PALS and PACS were significantly reduced in patients with severe aortic regurgitation compared to patients with mild and moderate aortic regurgitation [184]. Salas-Pacheco J.L. et al. have shown that left atrial biomechanics are similar in both aortic regurgitation and stenosis; a larger left atrial volume usually predicts worse atrial strain values [186]. Furthermore, a one-unit decrease in atrial reservoir strain was associated with a 6% increase in the probability of pulmonary hypertension [186]. The optimal timing of surgery in asymptomatic aortic regurgitation is challenging [119]. Ana Garcia Martin et al. have shown that diastolic function indices, including PALS, are prognostic markers of adverse events in asymptomatic severe aortic regurgitation [187]. Similarly to aortic stenosis, plasma levels of natriuretic peptides also increase with the severity of aortic regurgitation [30,188].

### 4.4. Mitral Valve Stenosis

Mitral valve stenosis is rare; however, it remains common in developing countries due to rheumatic etiology. Mitral stenosis significantly increases left atrial pressure through upstream hemodynamic effects, with less impact on the left ventricle. This can lead to pulmonary hypertension and right ventricular dysfunction. Furthermore, rheumatic etiology promotes atrial dysfunction through atrial myocytolysis and inflammation, supplementary to upstream pressure increase [189]. Progressive left atrial dysfunction, dilatation, and atrial fibrosis increase the risk of atrial fibrillation, which can develop by up to 30–50% in patients with mitral stenosis [2,190,191,192]. Previous studies have revealed a correlation between left atrial dimensions and atrial fibrillation; however, left atrial dysfunction might occur at an earlier stage [191,193,194]. Stassen J. et al. have shown that PALS ≤ 21% is independently associated with new-onset atrial fibrillation in a population with mixed severity mitral stenosis, including 40% severe mitral stenosis patients [191]. In asymptomatic mild or moderate mitral stenosis, PALS ≤ 16–17% has been defined as the cut-off value for predicting new-onset atrial fibrillation [176,195]. Bouchahda N. et al. have confirmed a correlation between left atrial reservoir strain, New York Heart Association (NYHA) heart failure functional state, and mitral stenosis severity [196,197]. A recent study has found a strong association between PALS, high pulmonary artery systolic pressure, atrial fibrillation, and right ventricular dysfunction in a population with severe mitral stenosis [198]. PALS ≤ 7% showed a >80% accuracy in identifying patients with severe hypertension and atrial fibrillation in the population with mitral stenosis [198]. Despite symptoms, the levels of natriuretic peptides might initially be lower compared to other valve diseases due to a cause upstream from the left ventricle and the consequent absence of left ventricular wall stress [30]. Nonetheless, natriuretic peptides correlate with the severity of mitral stenosis and NYHA functional class in general [30,199].

Mitral valve commissurotomy is used to repair mitral valve stenosis. Shenthar J. et al. observed a significant improvement in left atrial reservoir strain from 11.23 ± 6.83% pre-commissurotomy to 16.80 ± 8.82% at 6 months post-procedure time [189]. In addition, atrial fibrillation patients in this study had lower left atrial, left ventricular, and right ventricular strain values than patients with sinus rhythm [189]. Histopathological studies have revealed larger hypertrophied atrial cells with greater fibrosis and cellular degeneration in the mitral stenosis group undergoing mitral valve surgery than coronary artery bypass surgery patients [193]. Matrix metalloproteinases (MMP)-1 and MMP-9 were down-regulated in mitral stenosis patients, concordant with the extent of fibrosis, but regardless of arrhythmia [193].

### 4.5. Tricuspid Valve Disease

Tricuspid valve regurgitation is common in the general population [200]. The age- and sex-adjusted prevalence of moderate/severe tricuspid regurgitation is 0.55%, similar to the prevalence of aortic stenosis [201,202]. Tricuspid regurgitation is classified into primary (organic), secondary (functional), and cardiac implantable electronic device-related regurgitation [3]. The predominant mechanism of tricuspid regurgitation is functional with non-leaflet pathology [3]. It is the consequence of tricuspid annulus enlargement due to either right atrial enlargement (atrial functional tricuspid regurgitation) or right ventricular enlargement (ventricular functional tricuspid regurgitation) [3,203]. However, the anatomic features of the right atrium, tricuspid annulus, and right ventricle differ for atrial and ventricular functional phenotypes. The right atrium is more dilated, and the enlargement of the tricuspid annulus is more pronounced in atrial functional tricuspid regurgitation compared to the ventricular type [204,205,206]. Utsunomiya H. et al. have demonstrated that the right atrial volume index is 104 mL/m^2^ vs. 62 mL/m^2^, and the right atrial/right ventricular end-systolic volume ratio is 2.2 vs. 0.9 in atrial and ventricular phenotypes, respectively [206]. The most common form of tricuspid regurgitation is the ventricular functional type. This is the result of left-sided heart disease or right ventricular dysfunction leading to right ventricular pressure overload, which has a worse outcome compared to other etiologies [204,205]. Atrial functional tricuspid regurgitation is associated with right atrial dilatation and permanent atrial fibrillation, with an overall better outcome compared to the ventricular phenotype [207]. It is recognized that prolonged atrial fibrillation is associated with atrial remodeling, which might lead to atrial functional tricuspid regurgitation, resulting in a vicious cycle of both atrial cardiomyopathy and progression to tricuspid regurgitation [207,208,209]. Overall, atrial cardiomyopathy can be observed in all etiologies of tricuspid regurgitation. However, due to the long neglect of the right side of the heart, there are only few data on right atrial remodeling associated with tricuspid regurgitation [205]. Guta A.C. et al. found that the minimum right atrial volume was correlated to the severity of functional tricuspid regurgitation [207]. Furthermore, Wright L.M. et al. revealed that right atrial strain was correlated with right atrial size, right ventricular function, and inferior vena cava size [28] (Figure 5). Interestingly, there is a correlation between functional tricuspid regurgitation and left atrial reservoir strain in left heart disease etiology, such as mitral stenosis [210]. Furthermore, left atrial strain was predictive of recurrent tricuspid regurgitation after tricuspid valve annuloplasty [210]. Teixeira et al. found that tricuspid regurgitation and the consequent chronic volume overload state, atrial fibrillation, and right ventricular systolic longitudinal function significantly alter right atrial reservoir function [211].

## 5. Pharmacological Prevention in Atrial Cardiomyopathy

Sodium-glucose cotransporter 2 (SGLT2) inhibitors induce glucosuria, natriuresis, and osmotic diuresis leading to a reduction in cardiac preload, afterload, and myocardial oxygen consumption; subsequently, these drugs can potentially reduce atrial pressure and dilatation [212]. Recent, large, multicenter, randomized controlled trials have revealed that SGLT2 inhibitors significantly reduce hospitalization for heart failure and cardiovascular mortality [213,214,215]. A large meta-analysis involving 63,604 patients suggested that SGLT2 inhibitors significantly reduced the risk of atrial fibrillation [216]. The protective mechanism of SGLT2 inhibitors on the heart involves many mechanisms. Importantly, SGLT2 inhibitors can inhibit cardiomyocyte apoptosis and reduce myocardial fibrosis and adverse remodeling by alleviating oxidative stress, TGF-β production, and regulating macrophage polarization [217,218,219,220]. Furthermore, SGLT2 inhibitors suppress the natrium–hydrogen exchanger in cardiomyocytes, leading to a reduction in the intracellular sodium content, which subsequently results in a lower activity of the natrium/calcium exchanger, and a decrease in the sarcoplasmic reticulum calcium levels; hence, the risk of arrhythmia decreases [217]. It is presumed that epicardial fat could increase the risk of atrial fibrillation via localized inflammation [221,222]. Nonetheless, recent evidence has suggested that SGLT2 inhibitors can reduce the thickness and volume of epicardial adipose tissue by promoting fat burning and normalizing the lipogenesis-to-lipolysis ratio [223,224,225,226]. Moreover, SGLT2 inhibitors can also reduce lipid peroxidation and oxidative damage, and modulate epicardial adipocyte differentiation [225,227].

Atrial fibrosis and atrial fibrillation might be initiated by aldosterone binding to the mineralocorticoid receptor and promoting inflammation, oxidative stress, and connective tissue growth factor upregulation [228,229]. Previous studies have revealed that aldosterone pathway blockade with mineralocorticoid receptor antagonists (MRA) reduces atrial fibrosis and the risk of new-onset and recurrent atrial fibrillation [230,231]. Experimental data have revealed that MRAs suppress oxidative stress and fibrosis independently of their blood pressure lowering effects [229]. However, Pretorius et al. demonstrated that MRAs did not significantly reduce the risk of postoperative atrial fibrillation [232]. Consequently, MRAs may not suppress the immediate inflammatory response in postoperative atrial fibrillation but could reduce the fibrosis and atrial fibrillation burden [229,231].

## 6. Conclusions

In conclusion, atrial cardiomyopathy has been defined as structural, functional, or electrophysiological atrial changes with potential clinical manifestations [7]. Clinical data, such as serum biomarkers, ECG, and conventional and advanced imaging techniques, can provide information on the diagnosis, and the anatomical and functional features of atria and valves, as well as provide prognostic information for valvular heart disease [6]. These features of atrial cardiomyopathy may vary depending on the type and severity of valvular disease, representing different hemodynamic loads on the atria [2]. Some pathophysiological steps in atrial cardiomyopathy have already been explored, including cellular, molecular, neurohormonal, and biomechanical adaptive responses to valvular disease [5]. Previous histopathological studies have revealed atrial fibrosis in the background of atrial cardiomyopathy. Furthermore, the amount of atrial fibrosis may correlate with atrial function. Increasing evidence suggests that left atrial function, defined by speckle tracking echocardiography-derived strain parameters, can provide important prognostic information in valvular diseases. In addition, atrial dysfunction may precede atrial enlargement and predict the development of atrial fibrillation and the risk of major adverse cardiovascular events. However, questions about atrial cardiomyopathy still remain unanswered in both bedside and bench-side research. Besides the known interactions between atrial cardiomyopathy and valvular heart disease, a more-in-depth cellular and clinical characterization of atrial cardiomyopathy is lacking. There is a paucity of histopathological and molecular biological data on the different stages of atrial cardiomyopathy. Moreover, a deeper pathophysiological understanding and biological approach would help in developing new drug innovations to delay disease progression. From a clinical perspective, the most important question is the usefulness of atrial measurements in clinical decision-making. It is essential to select the most appropriate atrial markers to detect the different stages of atrial cardiomyopathy and to define prognostic cut-off points in order to apply them in clinically meaningful management strategies. Nonetheless, there is no accepted prognostic atrial strain cut-off value in many clinical settings. The point of irreversible progression is still not known, when irreversible maladaptive atrial deterioration develops regardless of treatment. Despite advances in the clinical and cellular characterization of atrial cardiomyopathy, further comprehensive work is needed to improve risk assessment, diagnostic accuracy, and patient selection for optimal valvular disease management.

## Figures and Tables

**Figure 1 cells-12-01796-f001:**
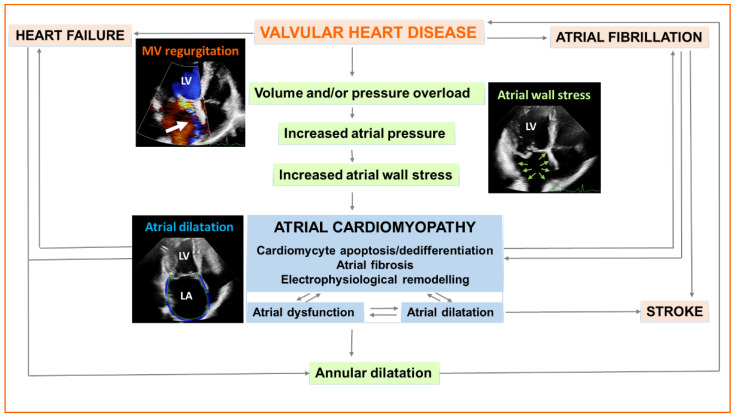
Schematic illustration of atrial cardiomyopathy in valvular heart disease. Valvular heart disease, such as primary (organic) mitral valve (MV) regurgitation due to leaflet prolapse (white arrow), leads to atrial volume and/or pressure overload and consequent increased atrial wall stress (green arrows), which results in cardiomyocyte apoptosis, cardiomyocyte dedifferentiation, and atrial fibrosis. Atrial dysfunction and/or atrial dilatation with electrophysiological remodeling develops defined as atrial cardiomyopathy. Atrial cardiomyopathy leads to annular dilatation and exaggerates valve malcoaptation. Atrial fibrillation might develop and increase the risk of stroke. Notably, longstanding permanent atrial fibrillation without initial valvular heart disease may lead to atrial dilatation, subsequent annular dilatation, and secondary (functional) mitral or tricuspid valve regurgitation. Aortic stenosis may also promote atrial cardiomyopathy due to pressure overload. In advanced stages of aortic stenosis, functional mitral or tricuspid regurgitation can be detected due to mitral or tricuspid annulus dilatation. Heart failure symptoms develop when the adaptive mechanisms fail to further compensate the pathophysiological changes. LA: left atrium; LV: left ventricle.

**Figure 2 cells-12-01796-f002:**
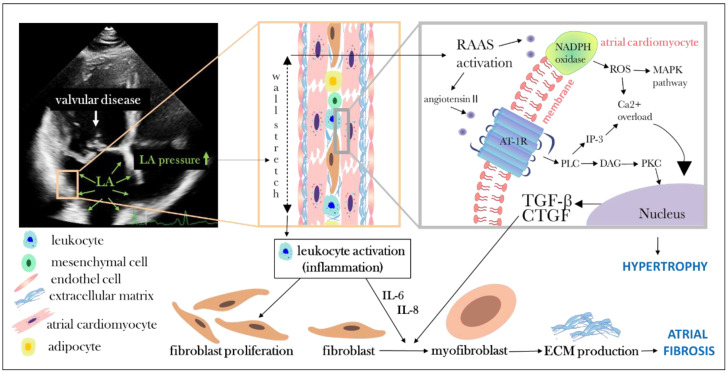
Schematic illustration of cellular and molecular changes in valvular disease leading to atrial fibrosis and cardiomyocyte hypertrophy. Valvular disease leads to increased left atrial (LA) pressure and wall stretch resulting in renin–angiotensin–aldosterone system (RAAS) and leukocyte activation, which promote atrial fibrosis. NADPH: nicotinamide adenine dinucleotide phosphate; ROS: reactive oxygen species; MAPK: mitogen-activated protein kinase; Ca: calcium; AT-1R: angiotensin type 1 receptor; PLC: phospholipase C; IP3: inositol triphosphate 3; DAG: diacylglycerol; PKC: protein kinase C; TGF-β: transforming growth factor β; CTGF: connective tissue growth factor; ECM: extracellular matrix.

**Figure 3 cells-12-01796-f003:**
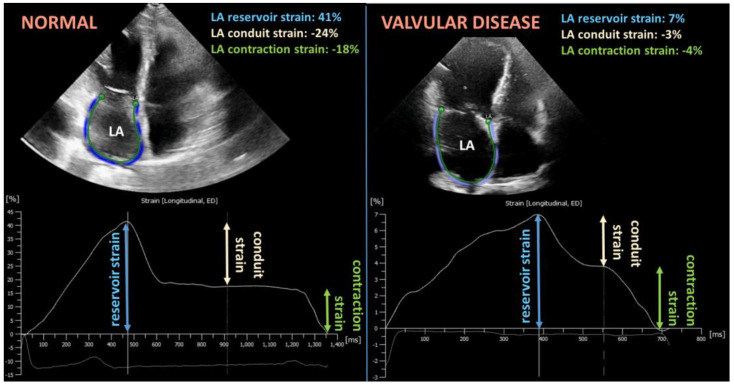
Representative image of left atrial strain analysis using two dimensional speckle tracking echocardiography in case of a normal subject (**left panel**) and valvular heart disease patient (**right panel**). The absolute values of reservoir, conduit, and contraction strain parameters are significantly decreased in the valvular heart disease case, suggesting left atrial dysfunction. LA: left atrium.

**Figure 4 cells-12-01796-f004:**
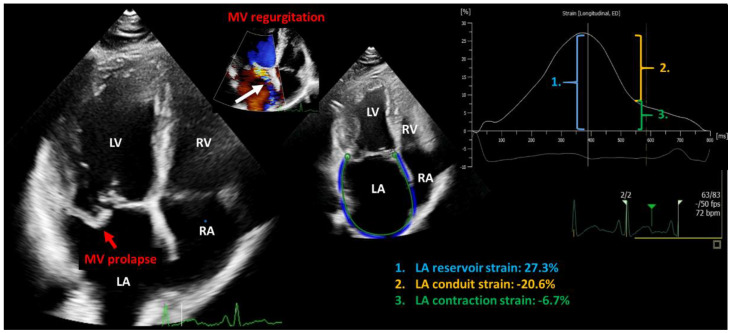
Representative two-dimensional transthoracic echocardiography image showing primary mitral regurgitation due to prolapse of posterior mitral valve leaflet (red arrow) leading to eccentric mitral regurgitation (white arrow). The left atrial reservoir function is decreased (1.) LA: left atrium; RA: right atrium; LV: left ventricle; RV: right ventricle; MV: mitral valve; (2.) conduit strain; (3.) contraction strain.

**Figure 5 cells-12-01796-f005:**
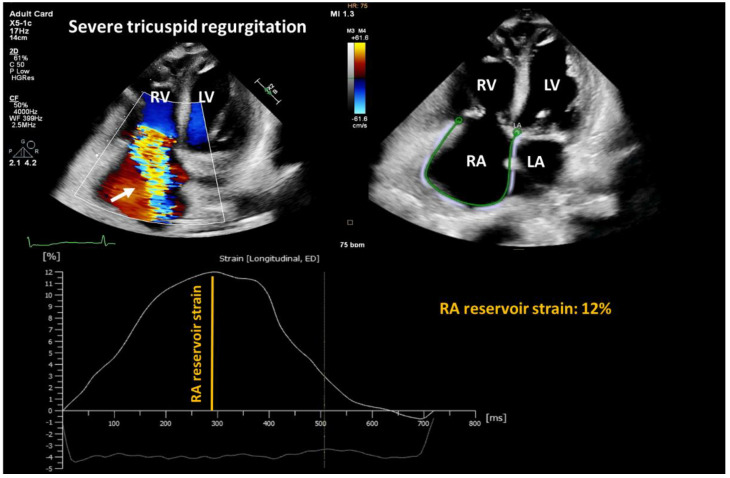
Representative two-dimensional transthoracic echocardiography image showing severe tricuspid regurgitation (white arrow) and decreased right atrial reservoir function. RA: right atrium; LA: left atrium; RV: right ventricle; LV: left ventricle.

## Data Availability

Not applicable.

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
