# Peer review of "Atrial Cardiomyopathy in Valvular Heart Disease: From Molecular Biology to Clinical Perspectives"

_cells, 2023, doi:10.3390/cells12131796_

Round 1
Reviewer 1 Report
In chapter Imaging markers... Transthoracic echocardiography ...It will be usefull to describe not only left atrial changes, but also interatrial septum degeneration and deformation including PFO. Many degenerations of the membranose part of IAS is easy detectible by echocardiography
Author Response
Response to Reviewer#1
We would like to express our thanks to Reviewer#1 for the careful evaluation of the manuscript and the helpful and constructive suggestions. We have heeded the Reviewer’s helpful propositions and prepared a revised version of the manuscript, which includes the alterations suggested by the Reviewer. Please find our response to the comments below.
Reviewer’s Comments and Suggestions for Authors:
“In chapter Imaging markers... Transthoracic echocardiography ...It will be usefull to describe not only left atrial changes, but also interatrial septum degeneration and deformation including PFO. Many degenerations of the membranose part of IAS is easy detectible by echocardiography”
Response to Reviewer#1: Thank You for the useful suggestion! Accordingly, we have modified the manuscript (highlighted in yellow, rows 302-316):
“The presence of spontaneous echocardiographic contrast in the left atrium and reduced left atrial appendage blood flow velocity were recognized as markers of atrial cardiomyopathy and associated with an increased risk of thrombus formation [80,91,92]. Furthermore, transthoracic and transesophageal echocardiography can easily detect deformations of the interatrial septum, such as interatrial septal aneurysm (IASA), patent foramen ovale (PFO) or atrial septal pouch (ASP) [93-95]. The IASA is defined as a saccular deformation of the atrial septum, the PFO is a flaplike opening between the atrial septum primum and secundum at the location of the fossa ovalis, and the diverticulum-like ASP occurs when the septum primum and secundum are incompletely fused [93,96]. These deformations are supposed to be associated with thromboembolic events in different mechanisms [93]. Local thrombus formation related to blood stasis within these deformations might represent one mechanism [93]. However, thromboembolism transit through the PFO channel might serve as a further potential mechanism, defined as a paradoxical embolism [93]. Notably, the prevalence of PFO is approximately 25% in the healthy adult population, nonetheless, the risk of embolization in this population is mainly associated with several clinical and morphological factors, such as the length of the PFO channel, the presence of concomitant IASA or prominent Eustachian valve and absence of cardiovascular risk factors [93,97,98].
Advanced echocardiography methods, such as speckle tracking echocardiography represent a sensitive tool to assess atrial function. This technique analyses the unique fingerprint-like natural acoustic marker pattern (or speckle pattern) of grayscale B-mode images of the myocardium, which is tracked consecutively frame-by-frame during the cardiac cycle [99].”
New references:
- Corban, M.T.; Toya, T.; Ahmad, A.; Lerman, L.O.; Lee, H.C.; Lerman, A. Atrial Fibrillation and
Endothelial Dysfunction: A Potential Link? Mayo Clin Proc 2021, 96, 1609-1621,
doi:10.1016/j.mayocp.2020.11.005.
- Guazzi, M.; Arena, R. Endothelial dysfunction and pathophysiological correlates in atrial fibrillation. Heart 2009, 95, 102-106, doi:10.1136/hrt.2007.135277.
- Takahashi, N.; Ishibashi, Y.; Shimada, T.; Sakane, T.; Ohata, S.; Sugamori, T.; Ohta, Y.; Inoue, S.; Nakamura, K.; Shimizu, H.; et al. Impaired exercise-induced vasodilatation in chronic atrial fibrillation--role of endothelium-derived nitric oxide. Circ J 2002, 66, 583-588, doi:10.1253/circj.66.583.
- Zheng, L.H.; Sun, W.; Yao, Y.; Hou, B.B.; Qiao, Y.; Zhang, S. Associations of big endothelin-1 and C-reactive protein in atrial fibrillation. J Geriatr Cardiol 2016, 13, 465-470, doi:10.11909/j.issn.1671-5411.2016.05.005.
- Dudkiewicz, D.; Hołda, M.K. Interatrial septum as a possible source of thromboembolic events. Translational Research in Anatomy 2022, 27, 100190, doi:https://fanyv88.com:443/https/doi.org/10.1016/j.tria.2022.100190.
- Molnár, A.; Ábrahám, P.; Merkely, B.; Nardai, S. Echocardiographic Evaluation of Atrial Communications before Transcatheter Closure. J Vis Exp 2022, doi:10.3791/61240.
- Silvestry, F.E.; Cohen, M.S.; Armsby, L.B.; Burkule, N.J.; Fleishman, C.E.; Hijazi, Z.M.; Lang, R.M.; Rome, J.J.; Wang, Y. Guidelines for the Echocardiographic Assessment of Atrial Septal Defect and Patent Foramen Ovale: From the American Society of Echocardiography and Society for Cardiac Angiography and Interventions. J Am Soc Echocardiogr 2015, 28, 910-958, doi:10.1016/j.echo.2015.05.015.
- Hołda, M.K.; Koziej, M.; Hołda, J.; Piątek, K.; Tyrak, K.; Chołopiak, W.; Bolechała, F.; Walocha, J.A.; Klimek-Piotrowska, W. Atrial septal pouch - Morphological features and clinical considerations. Int J Cardiol 2016, 220, 337-342, doi:10.1016/j.ijcard.2016.06.141.
- Hołda, M.K.; Krawczyk-Ożóg, A.; Koziej, M.; Kołodziejczyk, J.; Sorysz, D.; Szczepanek, E.; Jędras, J.; Dudek, D. Patent Foramen Ovale Channel Morphometric Characteristics Associated with Cryptogenic Stroke: The MorPFO Score. J Am Soc Echocardiogr 2021, 34, 1285-1293.e1283, doi:10.1016/j.echo.2021.07.016.
- Kleindorfer, D.O.; Towfighi, A.; Chaturvedi, S.; Cockroft, K.M.; Gutierrez, J.; Lombardi-Hill, D.; Kamel, H.; Kernan, W.N.; Kittner, S.J.; Leira, E.C.; et al. 2021 Guideline for the Prevention of Stroke in Patients With Stroke and Transient Ischemic Attack: A Guideline From the American Heart Association/American Stroke Association. Stroke 2021, 52, e364-e467, doi:10.1161/str.0000000000000375.
- Ni, L.; Yuan, C.; Chen, G.; Zhang, C.; Wu, X. SGLT2i: beyond the glucose-lowering effect. Cardiovasc Diabetol 2020, 19, 98, doi:10.1186/s12933-020-01071-y.
- McMurray, J.J.V.; Solomon, S.D.; Inzucchi, S.E.; Køber, L.; Kosiborod, M.N.; Martinez, F.A.; Ponikowski, P.; Sabatine, M.S.; Anand, I.S.; Bělohlávek, J.; et al. Dapagliflozin in Patients with Heart Failure and Reduced Ejection Fraction. N Engl J Med 2019, 381, 1995-2008, doi:10.1056/NEJMoa1911303.
- Packer, M.; Anker, S.D.; Butler, J.; Filippatos, G.; Pocock, S.J.; Carson, P.; Januzzi, J.; Verma, S.; Tsutsui, H.; Brueckmann, M.; et al. Cardiovascular and Renal Outcomes with Empagliflozin in Heart Failure. N Engl J Med 2020, 383, 1413-1424, doi:10.1056/NEJMoa2022190.
- Petrie, M.C.; Verma, S.; Docherty, K.F.; Inzucchi, S.E.; Anand, I.; Belohlávek, J.; Böhm, M.; Chiang, C.E.; Chopra, V.K.; de Boer, R.A.; et al. Effect of Dapagliflozin on Worsening Heart Failure and Cardiovascular Death in Patients With Heart Failure With and Without Diabetes. Jama 2020, 323, 1353-1368, doi:10.1001/jama.2020.1906.
- Zheng, R.J.; Wang, Y.; Tang, J.N.; Duan, J.Y.; Yuan, M.Y.; Zhang, J.Y. Association of SGLT2 Inhibitors With Risk of Atrial Fibrillation and Stroke in Patients With and Without Type 2 Diabetes: A Systemic Review and Meta-Analysis of Randomized Controlled Trials. J Cardiovasc Pharmacol 2022, 79, e145-e152, doi:10.1097/fjc.0000000000001183.
- Lee, T.M.; Chang, N.C.; Lin, S.Z. Dapagliflozin, a selective SGLT2 Inhibitor, attenuated cardiac fibrosis by regulating the macrophage polarization via STAT3 signaling in infarcted rat hearts. Free Radic Biol Med 2017, 104, 298-310, doi:10.1016/j.freeradbiomed.2017.01.035.
- Li, C.; Zhang, J.; Xue, M.; Li, X.; Han, F.; Liu, X.; Xu, L.; Lu, Y.; Cheng, Y.; Li, T.; et al. SGLT2 inhibition with empagliflozin attenuates myocardial oxidative stress and fibrosis in diabetic mice heart. Cardiovasc Diabetol 2019, 18, 15, doi:10.1186/s12933-019-0816-2.
- Nishinarita, R.; Niwano, S.; Niwano, H.; Nakamura, H.; Saito, D.; Sato, T.; Matsuura, G.; Arakawa, Y.; Kobayashi, S.; Shirakawa, Y.; et al. Canagliflozin Suppresses Atrial Remodeling in a Canine Atrial Fibrillation Model. J Am Heart Assoc 2021, 10, e017483, doi:10.1161/jaha.119.017483.
- Sun, H.Y.; Wang, N.P.; Halkos, M.E.; Kerendi, F.; Kin, H.; Wang, R.X.; Guyton, R.A.; Zhao, Z.Q. Involvement of Na+/H+ exchanger in hypoxia/re-oxygenation-induced neonatal rat cardiomyocyte apoptosis. Eur J Pharmacol 2004, 486, 121-131, doi:10.1016/j.ejphar.2003.12.016.
- Hatem, S.N.; Sanders, P. Epicardial adipose tissue and atrial fibrillation. Cardiovasc Res 2014, 102, 205-213, doi:10.1093/cvr/cvu045.
- Patel, K.H.K.; Hwang, T.; Se Liebers, C.; Ng, F.S. Epicardial adipose tissue as a mediator of cardiac arrhythmias. Am J Physiol Heart Circ Physiol 2022, 322, H129-h144, doi:10.1152/ajpheart.00565.2021.
- Gaborit, B.; Ancel, P.; Abdullah, A.E.; Maurice, F.; Abdesselam, I.; Calen, A.; Soghomonian, A.; Houssays, M.; Varlet, I.; Eisinger, M.; et al. Effect of empagliflozin on ectopic fat stores and myocardial energetics in type 2 diabetes: the EMPACEF study. Cardiovasc Diabetol 2021, 20, 57, doi:10.1186/s12933-021-01237-2.
- Masson, W.; Lavalle-Cobo, A.; Nogueira, J.P. Effect of SGLT2-Inhibitors on Epicardial Adipose Tissue: A Meta-Analysis. Cells 2021, 10, doi:10.3390/cells10082150.
- Yaribeygi, H.; Maleki, M.; Butler, A.E.; Jamialahmadi, T.; Sahebkar, A. Sodium-glucose co-transporter-2 inhibitors and epicardial adiposity. European Journal of Pharmaceutical Sciences 2023, 180, 106322, doi:https://fanyv88.com:443/https/doi.org/10.1016/j.ejps.2022.106322.
- Szekeres, Z.; Toth, K.; Szabados, E. The Effects of SGLT2 Inhibitors on Lipid Metabolism. Metabolites 2021, 11, doi:10.3390/metabo11020087.
- Llorens-Cebrià, C.; Molina-Van den Bosch, M.; Vergara, A.; Jacobs-Cachá, C.; Soler, M.J. Antioxidant Roles of SGLT2 Inhibitors in the Kidney. Biomolecules 2022, 12, doi:10.3390/biom12010143.
- Lavall, D.; Selzer, C.; Schuster, P.; Lenski, M.; Adam, O.; Schäfers, H.J.; Böhm, M.; Laufs, U. The mineralocorticoid receptor promotes fibrotic remodeling in atrial fibrillation. J Biol Chem 2014, 289, 6656-6668, doi:10.1074/jbc.M113.519256.
- Mayyas, F.; Alzoubi, K.H.; Van Wagoner, D.R. Impact of aldosterone antagonists on the substrate for atrial fibrillation: aldosterone promotes oxidative stress and atrial structural/electrical remodeling. Int J Cardiol 2013, 168, 5135-5142, doi:10.1016/j.ijcard.2013.08.022.
- Liu, T.; Korantzopoulos, P.; Shao, Q.; Zhang, Z.; Letsas, K.P.; Li, G. Mineralocorticoid receptor antagonists and atrial fibrillation: a meta-analysis. Europace 2016, 18, 672-678, doi:10.1093/europace/euv366.
- Neefs, J.; van den Berg, N.W.; Limpens, J.; Berger, W.R.; Boekholdt, S.M.; Sanders, P.; de Groot, J.R. Aldosterone Pathway Blockade to Prevent Atrial Fibrillation: A Systematic Review and Meta-Analysis. Int J Cardiol 2017, 231, 155-161, doi:10.1016/j.ijcard.2016.12.029.
- Pretorius, M.; Murray, K.T.; Yu, C.; Byrne, J.G.; Billings, F.T.t.; Petracek, M.R.; Greelish, J.P.; Hoff, S.J.; Ball, S.K.; Mishra, V.; et al. Angiotensin-converting enzyme inhibition or mineralocorticoid receptor blockade do not affect prevalence of atrial fibrillation in patients undergoing cardiac surgery. Crit Care Med 2012, 40, 2805-2812, doi:10.1097/CCM.0b013e31825b8be2.
Once again, we would like to thank the Reviewer for the insightful comments and suggestions! We believe these resulted in a much-improved manuscript that may be acceptable for publication.
Andrea Agnes Molnar, MD, PhD
corresponding author
Please see the attachment

Reviewer 2 Report
Atrial cardiomyopathy is defined as any complex of structural, architectural, contractile or electrophysiological changes affecting the atria with the potential to produce clinically-relevant manifestations. And so it has been actively studied in recent years. This review fully reflects the stated topic of the development of atrial cardiomyopathy in valvular heart disease. At the moment, it is generally accepted that atrial cardiomyopathy is any functional and histological changes in the atrial myocardium. This review fills a gap in our knowledge of the development of atrial cardiomyopathy in valvular heart disease.
Minor edits
line 9 - remove the hyphen.
Author Response
Response to Reviewer2#
We would like to thank Reviewer#2 for the careful evaluation of the manuscript and helpful suggestions. Please find our response below.
Reviewer’s Comments and Suggestions for Authors:
“Atrial cardiomyopathy is defined as any complex of structural, architectural, contractile or electrophysiological changes affecting the atria with the potential to produce clinically-relevant manifestations. And so it has been actively studied in recent years. This review fully reflects the stated topic of the development of atrial cardiomyopathy in valvular heart disease. At the moment, it is generally accepted that atrial cardiomyopathy is any functional and histological changes in the atrial myocardium. This review fills a gap in our knowledge of the development of atrial cardiomyopathy in valvular heart disease.
Minor edits
line 9 - remove the hyphen.”
Response to Reviewer#2: Thank You for the valuable suggestion! Accordingly, we have modified the manuscript (highlighted in yellow, rows 9):
“Abstract: The review discusses the evolving topic of atrial cardiomyopathy concerning valvular heart disease. The pathogenesis of atrial cardiomyopathy involves multiple factors, such as valvular disease leading to atrial structural and functional remodeling due to pressure and volume overload.”
Andrea Agnes Molnar, MD, PhD
corresponding author
Please see the attachment.

Reviewer 3 Report
Your review of atrial cardiomyopathy in valvular heart disease and its role in AF is well written and of interest. You have focused on the role of atrial fibrosis-yo9u might also wish to review the role of atrial endothelial dysfunction in the pathophysiology of atrial cardiomyopathy. . Furthermore several pharmacologic interventions have been shown to prevent
AF in HF such as SGLT2is and MRAs . Some further review of the pharmacologic interventions that have affected the development of AF and their effect on atrial structure and function would be of value
Author Response
Response to Reviewer#3
We would like to thank Reviewer#3 for the careful evaluation of the manuscript and the helpful and constructive suggestions. We have heeded the Reviewer’s helpful propositions and prepared a revised version of the manuscript, which includes the alterations suggested by the Reviewer. Please find our response to the comments below.
Reviewer’s Comments and Suggestions for Authors:
“Your review of atrial cardiomyopathy in valvular heart disease and its role in AF is well written and of interest. You have focused on the role of atrial fibrosis-yo9u might also wish to review the role of atrial endothelial dysfunction in the pathophysiology of atrial cardiomyopathy. . Furthermore several pharmacologic interventions have been shown to prevent AF in HF such as SGLT2is and MRAs . Some further review of the pharmacologic interventions that have affected the development of AF and their effect on atrial structure and function would be of value”
Response to Reviewer#3: Thank You for the helpful suggestions! Accordingly, we have modified the manuscript (highlighted in yellow):
Rows 142-147:
“Notably, fibroblasts can connect to cardiomyocytes via gap-junctions and consequently act as passive electrical conduits; however, they are not electrically excitable [5,6,22]. A growing body of evidence supports that atrial fibrillation is associated with systemic vascular, coronary, and atrial endothelial dysfunction through multiple mechanisms [23]. Previous studies showed, that altered shear stress on endothelial cells, increased oxidative stress and inflammation, RAAS activation, intracellular Ca2+ overload, the release of endothelin-1 (ET-1), and reduced nitric oxide production, are the main pathways leading to atrial cardiomyopathy [23-26].
Functionally, the left atrium serves as a conduit between the pulmonary circulation and left ventricle [27].”
Rows 678-709:
“5. Pharmacological prevention in atrial cardiomyopathy
Sodium-glucose cotransporter 2 (SGLT2) inhibitors induce glucosuria, natriuresis, and osmotic diuresis leading to the reduction of cardiac preload, afterload, and myocardial oxygen consumption, subsequently, these drugs potentially can reduce atrial pressure and dilatation [212]. Recent, large, multicenter, randomized controlled trials revealed that SGLT2 inhibitors significantly reduce hospitalization for heart failure and cardiovascular mortality [213-215]. A large meta-analysis involving 63,604 patients suggested that SGLT2 inhibitors significantly reduced the risk of atrial fibrillation [216]. The protective mechanism of SGLT2 inhibitors on the heart involves many mechanisms. Importantly, SGLT2 inhibitors can inhibit cardiomyocyte apoptosis and reduce myocardial fibrosis and adverse remodeling by alleviating oxidative stress, TGF-β production, and regulating macrophage polarization [217-220]. Furthermore, SGLT2 inhibitors suppress natrium–hydrogen exchanger in cardiomyocytes, leading to a reduction in the intracellular sodium content, which subsequently results in a lower activity of the natrium/calcium exchanger and a decrease in the sarcoplasmic reticulum calcium levels, hence the risk of arrhythmia decreases [217]. It is presumed, that epicardial fat could increase the risk of atrial fibrillation by localized inflammation [221,222]. Nonetheless, recent evidence suggested, that SGLT2 inhibitors can reduce the thickness and volume of epicardial adipose tissue by promoting fat burning and normalizing the lipogenesis-to-lipolysis ratio [223-226]. Moreover, SGLT2 inhibitors can also reduce lipid peroxidation and oxidative damage, and modulate epicardial adipocyte differentiation [225,227].
Atrial fibrosis and atrial fibrillation might be initiated by aldosterone binding to the mineralocorticoid receptor and promoting inflammation, oxidative stress, and connective tissue growth factor upregulation[228,229]. Previous studies revealed, that aldosterone pathway blockade with mineralocorticoid receptor antagonists (MRA) reduces atrial fibrosis and the risk of new-onset and recurrent atrial fibrillation [230,231]. Experimental data revealed, that MRAs suppress oxidative stress and fibrosis independently of blood pressure lowering [229]. However, Pretorius and coworkers demonstrated, that MRAs did not significantly reduce the risk of postoperative atrial fibrillation [232]. Consequently, MRAs may not suppress the immediate inflammatory response in postoperative atrial fibrillation but could reduce fibrosis and atrial fibrillation burden [229,231].”
New references:
- Corban, M.T.; Toya, T.; Ahmad, A.; Lerman, L.O.; Lee, H.C.; Lerman, A. Atrial Fibrillation and
Endothelial Dysfunction: A Potential Link? Mayo Clin Proc 2021, 96, 1609-1621,
doi:10.1016/j.mayocp.2020.11.005.
- Guazzi, M.; Arena, R. Endothelial dysfunction and pathophysiological correlates in atrial fibrillation. Heart 2009, 95, 102-106, doi:10.1136/hrt.2007.135277.
- Takahashi, N.; Ishibashi, Y.; Shimada, T.; Sakane, T.; Ohata, S.; Sugamori, T.; Ohta, Y.; Inoue, S.; Nakamura, K.; Shimizu, H.; et al. Impaired exercise-induced vasodilatation in chronic atrial fibrillation--role of endothelium-derived nitric oxide. Circ J 2002, 66, 583-588, doi:10.1253/circj.66.583.
- Zheng, L.H.; Sun, W.; Yao, Y.; Hou, B.B.; Qiao, Y.; Zhang, S. Associations of big endothelin-1 and C-reactive protein in atrial fibrillation. J Geriatr Cardiol 2016, 13, 465-470, doi:10.11909/j.issn.1671-5411.2016.05.005.
- Dudkiewicz, D.; Hołda, M.K. Interatrial septum as a possible source of thromboembolic events. Translational Research in Anatomy 2022, 27, 100190, doi:https://fanyv88.com:443/https/doi.org/10.1016/j.tria.2022.100190.
- Molnár, A.; Ábrahám, P.; Merkely, B.; Nardai, S. Echocardiographic Evaluation of Atrial Communications before Transcatheter Closure. J Vis Exp 2022, doi:10.3791/61240.
- Silvestry, F.E.; Cohen, M.S.; Armsby, L.B.; Burkule, N.J.; Fleishman, C.E.; Hijazi, Z.M.; Lang, R.M.; Rome, J.J.; Wang, Y. Guidelines for the Echocardiographic Assessment of Atrial Septal Defect and Patent Foramen Ovale: From the American Society of Echocardiography and Society for Cardiac Angiography and Interventions. J Am Soc Echocardiogr 2015, 28, 910-958, doi:10.1016/j.echo.2015.05.015.
- Hołda, M.K.; Koziej, M.; Hołda, J.; Piątek, K.; Tyrak, K.; Chołopiak, W.; Bolechała, F.; Walocha, J.A.; Klimek-Piotrowska, W. Atrial septal pouch - Morphological features and clinical considerations. Int J Cardiol 2016, 220, 337-342, doi:10.1016/j.ijcard.2016.06.141.
- Hołda, M.K.; Krawczyk-Ożóg, A.; Koziej, M.; Kołodziejczyk, J.; Sorysz, D.; Szczepanek, E.; Jędras, J.; Dudek, D. Patent Foramen Ovale Channel Morphometric Characteristics Associated with Cryptogenic Stroke: The MorPFO Score. J Am Soc Echocardiogr 2021, 34, 1285-1293.e1283, doi:10.1016/j.echo.2021.07.016.
- Kleindorfer, D.O.; Towfighi, A.; Chaturvedi, S.; Cockroft, K.M.; Gutierrez, J.; Lombardi-Hill, D.; Kamel, H.; Kernan, W.N.; Kittner, S.J.; Leira, E.C.; et al. 2021 Guideline for the Prevention of Stroke in Patients With Stroke and Transient Ischemic Attack: A Guideline From the American Heart Association/American Stroke Association. Stroke 2021, 52, e364-e467, doi:10.1161/str.0000000000000375.
- Ni, L.; Yuan, C.; Chen, G.; Zhang, C.; Wu, X. SGLT2i: beyond the glucose-lowering effect. Cardiovasc Diabetol 2020, 19, 98, doi:10.1186/s12933-020-01071-y.
- McMurray, J.J.V.; Solomon, S.D.; Inzucchi, S.E.; Køber, L.; Kosiborod, M.N.; Martinez, F.A.; Ponikowski, P.; Sabatine, M.S.; Anand, I.S.; Bělohlávek, J.; et al. Dapagliflozin in Patients with Heart Failure and Reduced Ejection Fraction. N Engl J Med 2019, 381, 1995-2008, doi:10.1056/NEJMoa1911303.
- Packer, M.; Anker, S.D.; Butler, J.; Filippatos, G.; Pocock, S.J.; Carson, P.; Januzzi, J.; Verma, S.; Tsutsui, H.; Brueckmann, M.; et al. Cardiovascular and Renal Outcomes with Empagliflozin in Heart Failure. N Engl J Med 2020, 383, 1413-1424, doi:10.1056/NEJMoa2022190.
- Petrie, M.C.; Verma, S.; Docherty, K.F.; Inzucchi, S.E.; Anand, I.; Belohlávek, J.; Böhm, M.; Chiang, C.E.; Chopra, V.K.; de Boer, R.A.; et al. Effect of Dapagliflozin on Worsening Heart Failure and Cardiovascular Death in Patients With Heart Failure With and Without Diabetes. Jama 2020, 323, 1353-1368, doi:10.1001/jama.2020.1906.
- Zheng, R.J.; Wang, Y.; Tang, J.N.; Duan, J.Y.; Yuan, M.Y.; Zhang, J.Y. Association of SGLT2 Inhibitors With Risk of Atrial Fibrillation and Stroke in Patients With and Without Type 2 Diabetes: A Systemic Review and Meta-Analysis of Randomized Controlled Trials. J Cardiovasc Pharmacol 2022, 79, e145-e152, doi:10.1097/fjc.0000000000001183.
- Lee, T.M.; Chang, N.C.; Lin, S.Z. Dapagliflozin, a selective SGLT2 Inhibitor, attenuated cardiac fibrosis by regulating the macrophage polarization via STAT3 signaling in infarcted rat hearts. Free Radic Biol Med 2017, 104, 298-310, doi:10.1016/j.freeradbiomed.2017.01.035.
- Li, C.; Zhang, J.; Xue, M.; Li, X.; Han, F.; Liu, X.; Xu, L.; Lu, Y.; Cheng, Y.; Li, T.; et al. SGLT2 inhibition with empagliflozin attenuates myocardial oxidative stress and fibrosis in diabetic mice heart. Cardiovasc Diabetol 2019, 18, 15, doi:10.1186/s12933-019-0816-2.
- Nishinarita, R.; Niwano, S.; Niwano, H.; Nakamura, H.; Saito, D.; Sato, T.; Matsuura, G.; Arakawa, Y.; Kobayashi, S.; Shirakawa, Y.; et al. Canagliflozin Suppresses Atrial Remodeling in a Canine Atrial Fibrillation Model. J Am Heart Assoc 2021, 10, e017483, doi:10.1161/jaha.119.017483.
- Sun, H.Y.; Wang, N.P.; Halkos, M.E.; Kerendi, F.; Kin, H.; Wang, R.X.; Guyton, R.A.; Zhao, Z.Q. Involvement of Na+/H+ exchanger in hypoxia/re-oxygenation-induced neonatal rat cardiomyocyte apoptosis. Eur J Pharmacol 2004, 486, 121-131, doi:10.1016/j.ejphar.2003.12.016.
- Hatem, S.N.; Sanders, P. Epicardial adipose tissue and atrial fibrillation. Cardiovasc Res 2014, 102, 205-213, doi:10.1093/cvr/cvu045.
- Patel, K.H.K.; Hwang, T.; Se Liebers, C.; Ng, F.S. Epicardial adipose tissue as a mediator of cardiac arrhythmias. Am J Physiol Heart Circ Physiol 2022, 322, H129-h144, doi:10.1152/ajpheart.00565.2021.
- Gaborit, B.; Ancel, P.; Abdullah, A.E.; Maurice, F.; Abdesselam, I.; Calen, A.; Soghomonian, A.; Houssays, M.; Varlet, I.; Eisinger, M.; et al. Effect of empagliflozin on ectopic fat stores and myocardial energetics in type 2 diabetes: the EMPACEF study. Cardiovasc Diabetol 2021, 20, 57, doi:10.1186/s12933-021-01237-2.
- Masson, W.; Lavalle-Cobo, A.; Nogueira, J.P. Effect of SGLT2-Inhibitors on Epicardial Adipose Tissue: A Meta-Analysis. Cells 2021, 10, doi:10.3390/cells10082150.
- Yaribeygi, H.; Maleki, M.; Butler, A.E.; Jamialahmadi, T.; Sahebkar, A. Sodium-glucose co-transporter-2 inhibitors and epicardial adiposity. European Journal of Pharmaceutical Sciences 2023, 180, 106322, doi:https://fanyv88.com:443/https/doi.org/10.1016/j.ejps.2022.106322.
- Szekeres, Z.; Toth, K.; Szabados, E. The Effects of SGLT2 Inhibitors on Lipid Metabolism. Metabolites 2021, 11, doi:10.3390/metabo11020087.
- Llorens-Cebrià, C.; Molina-Van den Bosch, M.; Vergara, A.; Jacobs-Cachá, C.; Soler, M.J. Antioxidant Roles of SGLT2 Inhibitors in the Kidney. Biomolecules 2022, 12, doi:10.3390/biom12010143.
- Lavall, D.; Selzer, C.; Schuster, P.; Lenski, M.; Adam, O.; Schäfers, H.J.; Böhm, M.; Laufs, U. The mineralocorticoid receptor promotes fibrotic remodeling in atrial fibrillation. J Biol Chem 2014, 289, 6656-6668, doi:10.1074/jbc.M113.519256.
- Mayyas, F.; Alzoubi, K.H.; Van Wagoner, D.R. Impact of aldosterone antagonists on the substrate for atrial fibrillation: aldosterone promotes oxidative stress and atrial structural/electrical remodeling. Int J Cardiol 2013, 168, 5135-5142, doi:10.1016/j.ijcard.2013.08.022.
- Liu, T.; Korantzopoulos, P.; Shao, Q.; Zhang, Z.; Letsas, K.P.; Li, G. Mineralocorticoid receptor antagonists and atrial fibrillation: a meta-analysis. Europace 2016, 18, 672-678, doi:10.1093/europace/euv366.
- Neefs, J.; van den Berg, N.W.; Limpens, J.; Berger, W.R.; Boekholdt, S.M.; Sanders, P.; de Groot, J.R. Aldosterone Pathway Blockade to Prevent Atrial Fibrillation: A Systematic Review and Meta-Analysis. Int J Cardiol 2017, 231, 155-161, doi:10.1016/j.ijcard.2016.12.029.
- Pretorius, M.; Murray, K.T.; Yu, C.; Byrne, J.G.; Billings, F.T.t.; Petracek, M.R.; Greelish, J.P.; Hoff, S.J.; Ball, S.K.; Mishra, V.; et al. Angiotensin-converting enzyme inhibition or mineralocorticoid receptor blockade do not affect prevalence of atrial fibrillation in patients undergoing cardiac surgery. Crit Care Med 2012, 40, 2805-2812, doi:10.1097/CCM.0b013e31825b8be2.
Once again, we would like to thank the Reviewer for the insightful comments and suggestions! We do believe these resulted in a much-improved manuscript that may be acceptable for publication.
Andrea Agnes Molnar, MD, PhD
corresponding author
Please see the attachment.

Round 2
Reviewer 3 Report
The authors have responded to my comments and in my opinion the paper is now acceptable for publication.